# Research on Reward and Punishment Contract Model and Coordination of Green Supply Chain Based on Fairness Preference

**Mingjun Jiang [1,2,*], Dongyan Chen [3] and Hui Yu [3]**

1    School of Economics and Management, Harbin University of Science and Technology, Harbin 150080, China
2    School of Economics, Heilongjiang University of Science and Technology, Harbin 150022, China
3    College of Science, Harbin University of Science and Technology, Harbin 150080, China;
     chendonyan@hrbust.edu.cn (D.C.); hyu_hust@163.com (H.Y.)
*    Correspondence: junzi_2008@163.com

**Abstract:** With the increasing demand for "green" goods, it is necessary for companies to develop green innovation to seize market opportunities. Companies often use the model of supply chain cooperation to carry out green innovation. The standard reward and punishment contract model is constructed based on the green degree of the product provided by the supplier when the manufacturer has a fair preference. The impact of the manufacturer's fairness preference on the green degree of the product, price, manufacturer's profit, supplier's profit, and overall profit when the product green degree standard provided by the supplier is greater or smaller than the manufacturer's demand standard is analyzed. The impact of the difference in channel power between manufacturers and suppliers is also analyzed on the overall profit of the green supply chain. The research results showed that when the manufacturer's attention to fairness is equal to the attention to self-interest, the overall profit of the green supply chain is the largest, the coordination of the supply chain can be achieved, and the difference in the channel power of the participants in the green supply chain has a significant impact on the overall profit, which is verified by numerical analysis.

**Keywords:** green supply chain; fairness preference; reward and punishment contract; green degree

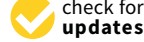



## 1. Introduction

As the major environmental burden and serious ecological problems worsen, the green and sustainable supply chain management mode has become the focus of attention all over the world. The green preference of consumers is increasing with the government and the attention of consumers to environmental protection. The implementation of green innovation and the production of green products by enterprises have become an inevitable trend for enterprises to seize market opportunities. Green innovation has been directly related to the business activities of enterprises, and green supply chain management is bound to become a new management model for the sustainable development of the supply chain. Strengthening green innovation is an effective strategy that not only solves environmental problems but also realizes economic development. From the perspective of consumers, more and more consumers are beginning to buy products with good environmental performance, such as environmental appliances, electric vehicles, ceramic tiles, ecological housing, solar energy, etc. Consumers' green consumption behaviors and various energy and resource crises have brought pressure to enterprises. Therefore, it is becoming more and more important for enterprises to pay more attention to environmental protection and increase the environmental protection of enterprise practice. Enterprises have to adapt to the new environment through innovation if they want to survive and develop, and enterprises need to constantly seek innovative solutions based on pressures from competitors, customers and regulators.

With people's attention to environmental issues, people pay more and more attention to the green supply chain, and the research of scholars on the green supply chain is more and more in-depth. There are many studies on Nash bargaining and supply chain coordination. Ye studies the coordination of a decentralized supply chain composed of suppliers and buyers, where the supplier only provided one product to the buyer, and the lead time can be controlled by increasing the rush cost. In addition, two supply chain inventory models with controllable lead times under different decision-making modes were studied: one is the decentralized inventory model based on Stackelberg model, and the other is the centralized inventory model based on integrated supply chain [1]. Sarkar formulated an integrated inventory model that allows the Stackelberg game policy for optimizing joint total cost of a vendor and buyer system [2]. Ye selected Guangdong, where the principal allocation method is the grandfathering approach as an illustrative case, and developed three preference cases including balanced weighting, eco-nomic-oriented weighting, and emission-oriented weighting [3]. Malik proposed a supply chain coordination method based on the lead-time crashing for a seller–buyer system wherein the seller motivates the buyer by reducing lead time to take part in coordinating decision making for the system's profit optimization [4]. Chu proposed new fixed cost allocation approaches for allocating a fixed cost among decision-making units with two-stage structures under the framework of data envelopment analysis. The proposed approaches always obtain a fixed cost allocation that is proportionally invariant [5]. Malik focused on the coordination between two players and cost-sharing in a supply chain management that considers a vendor and a buyer. For random demand and complex product production, a flexible production system is recommended [6].

Many scholars use the green degree to measure or represent the implementation degree of green innovation. Jiang constructed a green supply chain game model considering the green degree of products, and further constructed a green supply chain game model based on a revenue sharing contract through comparative analysis of equilibrium results [7]. Luo constructed a manufacturer-led, two-stage Stackelberg game model considering the green degree and studied the influence of government subsidy strategies on the decision making of participants in a closed-loop green supply chain and analyzed and compared the advantages and disadvantages of different subsidy strategies [8]. Wei studied the effects of partial heterogeneity and information asymmetry on the green degree of products and pricing in green supply chain. The results show that the asymmetry of preference information will reduce the green degree of the product and its wholesale price based on the analysis of the relationship between the green degree, the market demand and the product price under the condition of information asymmetry [9]. Xin constructed a manufacturer-led green supply chain decision-making model based on a shared contract and optimized the model by applying the Nash negotiation principle. Contract coordination conditions can improve the green degree of products [10]. Shang studied the impact of government subsidies on market demand and overall profit in retailer-led green supply chains and proposed that government subsidies have a positive effect on the green degree of products. When the market demand is more sensitive to the green degree of products, government subsidies are more beneficial to manufacturers when studying the green innovation of green supply chain [11]. Guan considered the disappointment avoidance behavior of members of the green supply chain, analyzed the influence of disappointment avoidance degree on the green degree of green supply chain by using the method of differential game and applied the cost sharing contract to coordinate the supply chain. He believed that the bidirectional cost sharing contract could realize the coordination of green supply chain [12].

Most studies on the green supply chain assume that all participants are completely rational. In fact, the influence of fairness preference of each member enterprise on the supply chain is inevitable, so some scholars also consider the fairness preference factor in the study of the green supply chain. Yang considered deferred payment and risk-free interest rates and analyzed the impact of manufacturers and retailers with fair preference on the green degree, the product price and the sales input of the green supply chain [13].

The green index and quality index are considered in the green supply chain model studied by Porkar. A set of Pareto optimal solutions are found by using the multi-objective optimization and genetic algorithm, and their practicability is proved [14]. Rong discussed the coordination problem with the goal of improving the green degree, as well as the impact of retailers' R&D costs on the green supply chain, and analyzed the revenue sharing realized by negotiation contracts in the decentralized situation [15]. Heydari constructed a two-stage green supply chain model and its coordination problem based on product sales price and product green degree. The product sales price is determined by the retailer and the manufacturer determines the product green degree. A hybrid contract of green cost sharing and revenue sharing is proposed [16]. In the reverse green supply chain, Yuan constructed a dynamic game model to analyze the green manufacturer's mixed recycling channel strategy and found that the green remanufacturer assumed environmental responsibility, and the recycler showed a strong fairness concern for profits. They also analyzed the impact of green remanufacturers' environmental responsibilities, fairness preferences, recycling prices and remanufacturers' optimal decisions [17]. Wang studied a two-stage green supply chain composed of a shared manufacturer and two competitive retailers. The influence of the manufacturer Stackelberg game, the retailer Stackelberg game and the vertical Nash game on the operation decision of supply chain members was discussed. The influence of consumers' green preferences and the green cost on the green supply chain decisions and profits was analyzed [18]. Habib studied the impact of strategic orientation in three dimensions, such as the green entrepreneurial orientation, the market orientation and the knowledge management orientation on the implementation of green supply chain management practices and the subsequently sustainable firm performance [19].

Green innovation is the core of the green supply chain. Green supply chain management and green innovation have strategic interconnections in developing new green products. Green supply chain management leads to green innovation [20]. The supplier contributes significant benefits to the environmental performance and competitive advantage of the firm through green innovation [21]. Zhang addressed the channel coordination problem in a green supply chain consisting of a manufacturer and a retailer, in which the manufacturer controls green innovation and wholesale price, while the retailer controls the sales price. Pricing and green innovation strategies in integrated and decentralized channels were computed and compared, and a two-part tariff contract was designed to coordinate the decentralized supply chain [22]. Noor assessed the level of adoption of green supply chain management and green innovation practices, and a total of 123 responses from Malaysian manufacturing companies were collected from mail questionnaire. They found that green supply chain management and green innovation practices are important to improve organisational environmental performance, which can directly offer great value [23]. Novitasari studied the assessment of green innovation as a mediating variable in the relationship between green supply chain management and firm performance. The sample collected by using purposive sampling method obtained 488 companies, assessed the mediation of green innovation in the relationship between green supply chain management and firm performance [24].

There is growing debate and interest in business cases on environmental sustainability and green innovation [25]. Despite the recent extensive research on green innovation achievements, the debate between researchers is still how to find the best way to implement such innovations [26]. Melander believes that through using the green supply chain to carry out strategic collaboration with partners, enterprises can fully obtain environmental protection materials, produce green environmental protection products, help enterprises to improve the innovation ability of green products and is beneficial to optimize the production process and green product innovation of enterprises [27]. The debate on green innovation best practices and how to implement these environmental goals still needs a common foundation. The best idea of green innovation should be easy to understand and, above all, be applied by all corporate stakeholders seeking better remedies for environmental degradation, economic growth and high energy consumption. Developing

environmental sustainability solutions and implementing green innovation are global challenges [28]. The increasing importance of green innovation is becoming a promising area of green supply chain management, where enterprises can eliminate the direct and indirect environmental impacts of the organization's final products [29]. Most Chinese enterprises have a low green innovation ability, and because they rely more on 'terminal' emission reduction, they cannot solve the problem of environmental pollution. Green innovation aims to avoid or reduce environmental damage through new or improved processes, technologies, systems and products [30]. Green innovation can solve the problem of environmental pollution from the source, and the concept of green innovation can support the implementation of green supply chain management by providing new ideas, methods or technologies for manufacturers to develop new products. Green innovation is considered to be able to continuously seek innovation methods at all stages of the supply chain in order to obtain competitive advantages and reduce environmental problems in industry [31]. Therefore, the concept of green innovation is the basis of a green supply chain management practice. It is supported by Lee and Kim, who claim that the basic innovation of supplier commitment in development of green products is to improve competitive advantage and environmental performance. Supplier commitment is one of the practices of implementing green supply chain management. It involves more green procurement, so suppliers are required to promise to provide manufacturers with materials to meet environmental requirements. Green innovation will then form the basis for this approach to develop new green products in a more strategic manner [32].

In the study of green supply chain considering fairness preference, Jiang established a profit game model of the green supply chain based on an F-S model considering fairness preference and analyzed the influence of the fairness preference of each participant in the green supply chain on the green degree and profit under the condition of asymmetric information [33]. Based on the retailer's fairness preference behaviors, Zhou studied the game model and decision-making problem of a dual-channel green supply chain and analyzed the influence of the fairness preference degree on the product's green degree and profit [34]. Gong constructed a green supply chain decision model considering corporate social responsibility and fairness preference, analyzed the influence of manufacturer's fairness preference on green degree and profit, and gave the optimal pricing decision, product green degree and overall profit of the supply chain [35]. Zhou proposed a new framework to explore fairness concern behavior through the survey of 225 manufacturing enterprises in China. The empirical analysis was carried out by a structural equation model, and the empirical results showed that knowledge sharing plays a key role in the green innovation of the supply chain [36]. Sang discussed the influence of reference price effect and fairness concern on green pricing and manufacturing strategy in three game models, the results showed that under the environment of considering the reference price effect and fairness concern, the green degree level, wholesale price and retail price of the product are decreased, and the retailer's profit is increased [37]. Li studied the impact of fairness concerns on product pricing and carbon emission reduction decisions in a two-stage green supply chain consisting of a fair neutral manufacturer and a retailer with fairness concerns [38].

From the current research results, most studies are based on the assumption that the members of the green supply chain are 'completely rational men'. Few studies consider the behavior characteristics of the members of the green supply chain, especially the fairness preferences. The 'complete rational man' only considers maximizing profits, but this assumption cannot explain the phenomenon in reality. Many studies need to further introduce the fairness preference problem to explain the phenomenon in reality. Many scholars have discussed the impact of fairness perception on contract signing, order quantity and pricing in terms of fairness preference and methods. Most of the supply chain studies are made by manufacturers and retailers, and some have considered the impact of fairness concerns on the green supply chain. There is an abundance of literature on green innovation, but there are few studies on the combination of green innovation and the sup-



ply chain, and the key role of green innovation cooperation in the sustainable development of enterprises is ignored. Most of the literature only focuses on the evaluation, selection and cooperation mechanism of supply chain partners, but the research on innovation investment decision making and the coordination of member enterprises in the green supply chain with green innovation investment as the main body is not deep enough, especially in terms of the problem of how to stimulate and improve the green degree of products from the perspective of fairness. Less studies consider the influence of manufacturers' reward and punishment mechanisms on green innovation investment decision making in the green supply chain, and the relationship between fairness preference and green degree, price and income is not studied in depth.

In the green supply chain, suppliers are the starting point. The green degree of raw materials provided by suppliers determines the green degree of final products. The green innovation of suppliers can reduce the energy consumption of suppliers, and the green innovation of suppliers can promote the ecological development of the whole supply chain. With the continuous development and reform of e-commerce, various e-commerce models have developed maturely. The competition and merger of the business in the interest chain, the integration of the upstream and downstream of the industrial chain, the resources will further move upward and, finally, the manufacturer will directly dock with the consumers to achieve 'de-retailer', which will be a trend of future supply chain development. Therefore, this paper studied a two-echelon supply chain system based on suppliers and manufacturers. A comparison between previous studies and this study is shown in Table 1.

**Table 1.** Contribution of the authors.

| Author(s) | Green Degree Improvement | Green Innovation | Channel Power in Supply Chain | Information Asymmetry | GSCM | Coordination of Supply Chain | Fairness Preference on Supply Chain |
|---|---|---|---|---|---|---|---|
| [1] | | | | | | ✓ | |
| [6] | | ✓ | | ✓ | ✓ | | |
| [8] | | ✓ | | | ✓ | ✓ | |
| [10] | | ✓ | | | ✓ | ✓ | |
| [13] | | ✓ | | | ✓ | | ✓ |
| [14] | ✓ | ✓ | | | ✓ | ✓ | |
| [18] | | ✓ | ✓ | ✓ | ✓ | | ✓ |
| [19] | | ✓ | ✓ | | ✓ | ✓ | |
| [22] | | ✓ | | | ✓ | | ✓ |
| [33] | ✓ | ✓ | | ✓ | ✓ | | ✓ |
| [34] | ✓ | ✓ | ✓ | | ✓ | | ✓ |
| [36] | | ✓ | | | | | ✓ |
| [38] | | | | | ✓ | | ✓ |
| This Paper | ✓ | ✓ | ✓ | ✓ | ✓ | ✓ | ✓ |

In the case of the manufacturers' fairness preference, a reward–penalty contract model based on the green degree of products, provided by suppliers as the reference standard, is constructed. Based on the 'ERC' fairness preference model, it is expanded and applied. The channel strength of each participant in the green supply chain is divided into two cases: equivalent and not equivalent. The influence of the improved reward–penalty contract based on the green degree of products on the supply chain coordination is analyzed. Finally, an example is used to verify the effectiveness of the proposed model.

## 2. Model Construction

### 2.1. Basic Assumptions

Based on the quantity-based reward–penalty contract proposed by Chiu [39], this paper improved the original model by using the green degree standard and constructed a

two-level green supply chain model composed of manufacturers and suppliers. Table 2 details the list of notations that are used in the paper.

**Table 2.** Notations table.

| Notations | Description | Notations | Description |
|---|---|---|---|
| $D$ | Market demand | $\mu$ | Penalty amount of unit product |
| $a$ | Market capacity | $\pi_m$ | Expected profit of the manufacturer |
| $p$ | Product price | $\pi_s$ | Expected profit of suppliers |
| $b$ | Consumer sensitivity coefficient to product price | $\pi(p,g)$ | Overall profit of green supply chain |
| $g$ | Product green degree | $p^*$ | Optimal product price |
| $\gamma$ | Consumer sensitivity coefficient to product green degree | $g^*$ | Optimal product green degree |
| $q$ | Manufacturer production | $\mu_m(\pi)$ | Utility of manufacturers under fairness preference |
| $b$ | Consumer sensitivity coefficient to product price | $\alpha$ | Level attention to its own income |
| $g$ | Product green degree | $\beta$ | Level of attention to fairness |
| $g_0$ | Manufacturer requires the supplier to provide product green degree | $p_1{}^*$ | Optimal product price when $g > g_0$ |
| $\delta$ | Coefficient of R&D cost | $g_1{}^*$ | Optimal product green degree when $g > g_0$ |
| $Cs$ | Supplier 's marginal production cost | $p_2{}^*$ | Optimal product price when $g < g_0$ |
| $Cm$ | Manufacturer 's marginal cost | $g_2{}^*$ | Optimal product green degree when $g < g_0$ |
| $T$ | Transfer payment | $p_3{}^*$ | Optimal product price when considering the different dominant position |
| $w$ | Wholesale price of green products provided by the supplier | $g_3{}^*$ | Optimal product green degree when considering the different dominant position |
| $\theta$ | Reward amount of unit product | $\varepsilon$ | Fair reference coefficient |

Make the following assumptions:

(1) Market demand $D$ is linear with product price and product green degree, $D = a - bp + \gamma g$.

(2) Manufacturer production equals market demand, i.e., $q = D$.

(3) In the green supply chain, the final green degree of the product is determined by the manufacturer. Assuming that the manufacturer requires the supplier to provide product green degree standard $g_0$, and the supplier provides product green degree $g$.

(4) The functional relationship between green innovation cost and green degree is $c(g) = \frac{1}{2}\delta g^2$, where $\delta$ is the coefficient of R&D cost, and the value of $\delta$ is a large normal number.

(5) When the green degree of the product provided by the supplier is $g \geq g_0$, the manufacturer rewards the supplier by increasing the unit price of the product, and $\theta$ is the reward amount of the unit product.

(6) When the green degree of the product provided by the supplier is $g < g_0$, the manufacturer punishes the supplier by reducing the unit price of the product, and $\mu$ is the penalty amount of the unit product.

(7) The supplier's marginal production cost is $C_s$, the manufacturer's marginal cost is $C_m$, the transfer payment $T$, the unit wholesale of green products provided by the supplier is w and the transfer payment from the manufacturer to the supplier is:

$$T = \begin{cases} (w + \theta)q, & g \geq g_0 \\ (w - \mu)q, & g < g_0 \end{cases} \tag{1}$$

*2.2. Model Analysis*

The expected profit of the manufacturer is $\pi_m$, calculated as:

$$\pi_m = pq - T - C_m q = (p - C_m)(a - bp + \gamma g) - T \tag{2}$$

The expected profit of the suppliers is $\pi_s$, calculated as:

$$\pi_s = T - C_s q - \frac{1}{2}\delta g^2 = T - C_s(a - bp + \gamma g) - \frac{1}{2}\delta g^2 \tag{3}$$

The overall profit of green supply chain is $\pi(p, g)$, calculated as:

$$\pi(p, g) = \pi_m + \pi_s = (p - C_m - C_s)(a - bp + \gamma g) - \frac{1}{2}\delta g^2 \tag{4}$$

The overall profit function $\pi(p, g)$ of the green supply chain is derived from the product price $p$ and the product green degree $g$, respectively, and we can obtain:

$$\frac{\partial \pi(p, g)}{\partial p} = a - 2bp + b(C_m + C_s) + \gamma g \tag{5}$$

$$\frac{\partial \pi(p, g)}{\partial g} = (p - C_m - C_s)\gamma - \delta g \tag{6}$$

The second derivatives of $\pi(p, g)$ with respect to $p$ and $g$ are obtained as:

$$\frac{\partial^2 \pi(p, g)}{\partial p^2} = -2b < 0 \tag{7}$$

$$\frac{\partial^2 \pi(p, g)}{\partial g^2} = -\delta < 0 \tag{8}$$

From Equations (5) and (7), it can be judged that the overall profit function $\pi(p, g)$ of the green supply chain is a concave function of the product price $p$, and from Equations (6) and (8), it can be seen that the overall profit function $\pi(p, g)$ is a concave function of the green degree $g$ of the product, which indicates that there is an optimal product price $p^*$ and an optimal green degree $g^*$. There is an optimal green innovation input for the green supply chain as a whole, and it is unique. Let $\frac{\partial \pi(p, g)}{\partial p} = 0$, $\frac{\partial \pi(p, g)}{\partial g} = 0$, and the simultaneous solution is:

$$p^* = \frac{(\gamma^2 - b\delta)(C_m + C_s) - a\delta}{\gamma^2 - 2b\delta} \tag{9}$$

$$g^* = \frac{b\gamma(C_m + C_s) - a\gamma}{\gamma^2 - 2b\delta} \tag{10}$$

## 3. Improved 'ERC' Model

The 'ERC' fairness preference model was first proposed by Bolton, which not only considers its own benefits, but also considers fairness, which is closer to reality. For the member enterprises in the green supply chain, they will also pay attention to the income of upstream and downstream enterprises while paying attention to their own income, which will promote the fairness of the income of the member enterprises in the green supply chain. 'ERC' Fairness Preference Theory focuses on the status of the enterprise itself in the supply chain. This theory assumes that among n members, the utility function of the $i$ ($i = 1, 2, 3 \dots n$) member is $u_i(x) = u_i(y_i, \sigma_i)$, where $y_i$ means positive returns for the whole, that is, $y_i \geq 0$, $\sigma_i = \sigma_i(y, c, n)$, where $c = \sum\limits_{i=1}^{n} y_i$, $y_i = c\sigma_i$, because $u_i(x)$ is differentiable and is a strictly increasing function, and when $\sigma_i = \frac{1}{n}$, the value is the largest [40]. This paper studies the game between suppliers and manufacturers in the green supply chain, that is, if $n = 2$, then $u_i(x) = u_i(y_i, \sigma_i) = u_i(c\sigma_i, \sigma_i) = \alpha_i c\sigma_i - \frac{1}{2}\beta_i\left(\sigma_i - \frac{1}{2}\right)^2$. Bolton pointed out that in the 'ERC' model, the game results of each member are affected

by the fairness preference of members, so the fairness preference factor is considered in the game between the two sides, and the utility function of the member enterprises is:

$$u_i = \alpha_i y_i - \frac{1}{2}\beta_i (y_i - \frac{c}{2})^2 \tag{11}$$

where $\alpha_i \geq 0$, $\beta_i \geq 0$, and the smaller $\alpha_i$ is, the more attention members pay to fairness. When $\alpha_i = 0$, it means that members of the supply chain are at an absolute fairness. The smaller $\beta_i$ is, the more attention members pay to their own interests. When $\beta_i = 0$, it means that members of the supply chain are absolute egoists, and the ratio of $\alpha_i/\beta_i$ is used to represent the degree of fairness preference of supply chain members.

In the green supply chain in which only manufacturers and suppliers are considered in this paper, it is assumed that manufacturers have a fair preference and the channel power of manufacturers and suppliers is equivalent, so the utility function of manufacturers can be further improved as follows:

$$\mu_m(\pi) = \alpha\pi_m - \beta\left(\pi_m - \frac{\pi}{2}\right) \tag{12}$$

$\alpha$ denotes the manufacturer's attention to its own income, $\beta$ denotes the attention to fairness and $\alpha/\beta$ denotes the manufacturer's fairness preference, where $\alpha > 0$, $\beta > 0$, and the manufacturer does not have the possibility of infinite pursuit of fairness, so $\alpha/\beta > 1/2$.

The manufacturer profit function $\pi_m$ and the green supply chain overall profit function $\pi(p, g)$ are substituted into (12), and the following results are obtained :

$$\mu_m(\pi) = (\alpha - \beta)\pi_m + \frac{\beta\pi}{2} = (\alpha - \beta)[(p - C_m)(a - bp + \gamma g) - T] + \frac{\beta}{2}\left[(p - C_m - C_s)(a - bp + \gamma g) - \frac{1}{2}\delta g^2\right] \tag{13}$$

## 4. Model Solving and Coordination Analysis

*4.1. The Product Green Degree Provided by the Supplier Is Greater Than Manufacturer Green Degree Standards*

When the green degree of the product provided by the supplier is greater than the green degree standard required by the manufacturer, the manufacturer rewards the supplier, and the utility function of the manufacturer is:

$$\mu_m(\pi) = (\alpha - \beta)\pi_m + \frac{\beta\pi}{2} = (\alpha - \beta)[(p - C_m - w - \theta)(a - bp + \gamma g)] + \frac{\beta}{2}[(p - C_m - C_s)(a - bp + \gamma g) - \frac{1}{2}\delta] \tag{14}$$

**Proposition 1.** *When the green degree of the product provided by the supplier is greater than the green degree standard of the manufacturer, namely $g > g_0$, there is only $p_1{}^*$, $g_1{}^*$.*

Such that $\frac{\partial \mu_m(\pi)}{\partial p} = 0$, $\frac{\partial \mu_m(\pi)}{\partial g} = 0$. Available:

$$p_1{}^* = \frac{2\gamma^2(\alpha - \beta)(C_m + w + \theta) + \beta\gamma^2(C_m + C_s) - \beta a\delta - \frac{2b\delta\beta}{2\alpha - \beta}\left[(\alpha - \beta)(C_m + w + \theta) + \frac{\beta}{2}(C_m + C_s)\right]}{(2\alpha - \beta)\gamma^2 - 2b\beta\delta} \tag{15}$$

$$g_1{}^* = \frac{2\gamma b(\alpha - \beta)(C_m + w + \theta) + b\gamma\beta(C_m + C_s) - \gamma a(2\alpha - \beta)}{(2\alpha - \beta)\gamma^2 - 2b\beta\delta} \tag{16}$$

**Proof of Proposition 1.** When $g > g_0$, the first-order and second-order derivatives of $p$ and $g$ are calculated for $\mu_m(\pi)$, respectively. It can be obtained that:

$$\frac{\partial \mu_m(\pi)}{\partial p} = (\alpha - \beta)[b(C_m + w + \theta) + a - 2bp + \gamma g] + \frac{\beta}{2}[(a - 2bp + \gamma g) + b(C_m + C_s)] \tag{17}$$

$$\frac{\partial \mu_m (\pi)}{\partial g} = \delta(\alpha - \beta)(p - C_m - w - \theta) + \frac{\beta}{2}[\gamma(p - C_m - C_s) - \delta g] \tag{18}$$

$$\frac{\partial^2 \mu_m (\pi)}{\partial p^2} = -b(2\alpha - \beta) \tag{19}$$

$$\frac{\partial^2 \mu_m (\pi)}{\partial g^2} = -\frac{\beta \delta}{2} \tag{20}$$

Because $\beta > 0$, $\frac{\alpha}{\beta} > \frac{1}{2}$, $\frac{\partial^2 \mu_m (\pi)}{\partial p^2} < 0$ and $\frac{\partial^2 \mu_m (\pi)}{\partial g^2} < 0$, then $\mu_m(\pi)$ is a strictly concave function, so there is only one optimal $p_1^*$, $g_1^*$. Let $\frac{\partial \mu_m (\pi)}{\partial p} = 0$ and $\frac{\partial \mu_m (\pi)}{\partial g} = 0$, which find the optimal value of the manufacturer's product price and green degree, and thus determine the green innovation investment. □

**Proposition 2.** *When $g > g_0$ and $\alpha = \beta$, the reward and punishment contract can coordinate the supply chain.*

**Proof of Proposition 2.** When $g > g_0$, the coordination of the green supply chain needs to meet the following conditions: $p_1^* = p^*$, $g_1^* = g^*$; that is:

$$\frac{2\gamma^2(\alpha - \beta)(C_m + w + \theta) + \beta\gamma^2(C_m + C_s) - \beta a\delta - \frac{2b\delta\beta}{2\alpha - \beta}\left[(\alpha - \beta)(C_m + w + \theta) + \frac{\beta}{2}(C_m + C_s)\right]}{(2\alpha - \beta)\gamma^2 - 2b\beta\delta} = \frac{(\gamma^2 - b\delta)(C_m + C_s) - a\delta}{\gamma^2 - 2b\delta} \tag{21}$$

$$\frac{2\gamma b(\alpha - \beta)(C_m + w + \theta) + b\gamma\beta(C_m + C_s) - \delta a(2\alpha - \beta)}{(2\alpha - \beta)\gamma^2 - 2b\beta\delta} = \frac{b\gamma(C_m + C_s) - a\gamma}{\gamma^2 - 2b\delta} \tag{22}$$

When $\alpha = \beta$, the left side of equation is simplified:

$$\frac{\beta\gamma^2(C_m + C_s) - \beta a\delta - \frac{2b\delta\beta}{2\alpha - \beta}\left[\frac{\beta}{2}(C_m + C_s)\right]}{(2\alpha - \beta)\gamma^2 - 2b\beta\delta} \tag{23}$$

Because $\beta > 0$, the numerator and denominator of the above formula are divided by $\beta$, and Equation (21) always holds. Similarly, when $\alpha = \beta$, the equality of (22) holds on both sides. Thus, we can determine that when $\alpha = \beta$, $p_1^* = p^*$, $g_1^* = g^*$, the reward and punishment contract can coordinate the green supply chain. □

**Property 1.** *When $g > g_0$, the manufacturer has a fair preference, and when $\alpha = \beta$, the overall profit of the green supply chain is the largest.*

**Proof of Property 1.** When $g > g_0$:

$$g_1^* = \frac{2\gamma b(\alpha - \beta)(C_m + w + \theta) + b\gamma\beta(C_m + C_s) - \gamma(2\alpha - \beta)}{(2\alpha - \beta)\gamma^2 - 2b\beta\delta} \tag{24}$$

Since $\beta > 0$, the molecular denominator of the above equation can be divided by $\beta$ at the same time, and the following equation can be obtained:

$$g_1^* = \frac{2\gamma b\left(\frac{\alpha}{\beta} - 1\right)(C_m + w + \theta) + b\gamma(C_m + C_s) - \gamma a\left(2\frac{\alpha}{\beta} - 1\right)}{\left(2\frac{\alpha}{\beta} - 1\right)\gamma^2 - 2b\delta} \tag{25}$$

Let $\frac{\alpha}{\beta} - 1 = x$, then $\frac{\alpha}{\beta} = x + 1$. The upper expression then becomes:

$$g_1^* = \frac{2\gamma bx(C_m + w + \theta) + b\gamma(C_m + C_s) - \gamma a(2x + 1)}{(2x + 1)\gamma^2 - 2b\delta} \tag{26}$$

Deriving the above equation from $x$, we can obtain:

$$\frac{dg_1{}^*}{dx} = \frac{2\gamma^3 b(w - C_s + \theta) + 4b\gamma\delta[a - b(w + C_m + \theta)]}{[(2x+1)\gamma^2 - 2b\delta]^2} > 0 \tag{27}$$

Since $\pi(p,g)$ is a concave function about $g$, and the maximum value is obtained when $g = g^*$, according to the superposition monotonicity judgment method of the two functions, the overall profit of the green supply chain increases as $\alpha/\beta$ increases. When $x > 0$, namely, $\alpha/\beta > 1$, the overall profit of the green supply chain decreases as $\alpha/\beta$ increases. When $\alpha = \beta$, the overall profit of the green supply chain is the largest. □

*4.2. Product Green Degree Provided by Suppliers Is Less Than Manufacturer Green Degree Standards*

When the green degree of the product provided by the supplier is less than the green degree required by the manufacturer, the manufacturer punishes the supplier, and the utility function of the manufacturer is:

$$\mu_m(\pi) = (\alpha - \beta)\pi_m + \frac{\beta\pi}{2} = (\alpha - \beta)[(p - C_m - w + \mu))(a - bp + \gamma g)] + \frac{\beta}{2}\left[(p - C_m - C_s)(a - bp + \gamma g) - \frac{1}{2}\delta g^2\right] \tag{28}$$

**Proposition 3.** *When the product green degree provided by the supplier is less than the manufacturer's green degree standard, that is, $g < g_0$, there is and only the unique $p_2{}^*$, $g_2{}^*$, available:*

$$p_2{}^* = \frac{2\gamma^2(\alpha - \beta)(C_m + w - \mu) + \beta\gamma^2(C_m + C_s) - \beta a\delta - \frac{2b\delta\beta}{2\alpha - \beta}\left[(\alpha - \beta)(C_m + w - \mu) + \frac{\beta}{2}(C_m + C_s)\right]}{(2\alpha - \beta)\gamma^2 - 2b\beta\delta} \tag{29}$$

$$g_2{}^* = \frac{2\gamma b(\alpha - \beta)(C_m + w - \mu) + b\gamma\beta(C_m + C_s) - \gamma a(2\alpha - \beta)}{(2\alpha - \beta)\gamma^2 - 2b\beta\delta} \tag{30}$$

**Proof of Proposition 3.** When $g < g_0$, the first-order and second-order derivatives of $p$ and $g$ are calculated for $\mu_m(\pi)$, respectively, and the following can be obtained:

$$\frac{\partial\mu_m(\pi)}{\partial p} = (\alpha - \beta)[b(C_m + w - \mu) + a - 2bp + \gamma g] + \frac{\beta}{2}[(a - 2bp + \gamma g) + b(C_m + C_s)] \tag{31}$$

$$\frac{\partial\mu_m(\pi)}{\partial g} = \delta(\alpha - \beta)(p - C_m - w + \mu) + \frac{\beta}{2}[\gamma(p - C_m - C_s) - \delta g] \tag{32}$$

$$\frac{\partial^2\mu_m(\pi)}{\partial p^2} = -b(2\alpha - \beta) \tag{33}$$

$$\frac{\partial^2\mu_m(\pi)}{\partial g^2} = -\frac{1}{2}\beta\delta \tag{34}$$

Since $\beta > 0$, $\frac{\alpha}{\beta} > \frac{1}{2}$, $\frac{\partial^2\mu_m(\pi)}{\partial p^2} < 0$, $\frac{\partial^2\mu_m(\pi)}{\partial g^2} < 0$, the function $\mu_m(\pi)$ is a strictly concave function, so there is and only the only optimal $p_2{}^*$, $g_2{}^*$. Let $\frac{\partial\mu_m(\pi)}{\partial p} = 0$, $\frac{\partial\mu_m(\pi)}{\partial g} = 0$. The optimal value of the manufacturer's product price and green degree can be obtained, thereby determining the investment in green innovation. □

**Proposition 4** . *When $g < g_0$, when the manufacturer pays equal attention to fairness and self-interest, that is, $\alpha = \beta$, the reward and punishment contract can coordinate the supply chain.*

**Proof of Proposition 4.** When $g < g_0$, the coordination of the green supply chain needs to meet the following conditions: $p_2^* = p^*$, $g_2^* = g^*$; that is:

$$\frac{2\gamma^2(\alpha - \beta)(C_m + w - \mu) + \beta\gamma^2(C_m + C_s) - \beta a\delta - \frac{2b\delta\beta}{2\alpha - \beta}\left[(\alpha - \beta)(C_m + w - \mu) + \frac{\beta}{2}(C_m + C_s)\right]}{(2\alpha - \beta)\gamma^2 - 2b\beta\delta} = \frac{(\gamma^2 - b\delta)(C_m + C_s) - a\delta}{\gamma^2 - 2b\delta} \tag{35}$$

$$\frac{2\gamma b(\alpha - \beta)(C_m + w - \mu) + b\gamma\beta(C_m + C_s) - \delta a(2\alpha - \beta)}{(2\alpha - \beta)\gamma^2 - 2b\beta\delta} = \frac{b\gamma(C_m + C_s) - a\gamma}{\gamma^2 - 2b\delta} \quad (36)$$

When $\alpha = \beta$, the left side of equation is simplified:

$$\frac{\beta\gamma^2(C_m + C_s) - \beta a\delta - \frac{2b\delta\beta}{2\alpha - \beta}\left[\frac{\beta}{2}(C_m + C_s)\right]}{(2\alpha - \beta)\gamma^2 - 2b\beta\delta} \quad (37)$$

Because $\beta > 0$, the upper molecule and denominator are divided by $\beta$, thus it can be concluded that the equality of (35) holds. Similarly, when $\alpha = \beta$, the equality of (36) holds on both sides. Thus, it can be determined that when $\alpha = \beta$, $p_2^* = p^*$, $g_2^* = g^*$, the reward and punishment contract mechanism can coordinate the green supply chain. □

**Property 2.** *When $g < g_0$, the manufacturer has a fair preference, and when $\alpha = \beta$, the overall profit of the green supply chain is the largest.*

**Proof of Property 2.** When $g < g_0$:

$$g_2^* = \frac{2\gamma b(\alpha - \beta)(C_m + w - \mu) + b\gamma\beta(C_m + C_s) - \gamma a(2\alpha - \beta)}{(2\alpha - \beta)\gamma^2 - 2b\beta\delta} \quad (38)$$

$\beta > 0$, and the upper molecular denominator can be divided by $\beta$ at the same time, as follows:

$$g_2^* = \frac{2\gamma b\left(\frac{\alpha}{\beta} - 1\right)(C_m + w - \mu) + b\gamma(C_m + C_s) - \gamma a\left(2\frac{\alpha}{\beta} - 1\right)}{\left(2\frac{\alpha}{\beta} - 1\right)\gamma^2 - 2b\delta} \quad (39)$$

Only when $\alpha = \beta$, the overall profit of the green supply chain is the largest, which proves that the process is the same as property 1. □

*4.3. Product Green Degree Provided by Suppliers Equals Manufacturer Green Degree Standards*

**Proposition 5.** *The green degree of products provided by suppliers is equal to the green degree standard of manufacturers. When and only when $\alpha = \beta$, the reward and punishment contract can coordinate the supply chain.*

It is proved that when $g_1^* \geq g_0 \geq g_2^*$, the green degree of the product provided by the manufacturer of the green supply chain is equal to the manufacturer's green degree standard reward standard $g_0$.

(1) First, assume $\alpha \neq \beta$. When $\alpha > \beta$, it is concluded that $g_1^* > g^*$ and $g_0 \geq g_1^*$, thus, $g_0 \geq g_1^* > g^*$. When $\alpha < \beta$, it is concluded that $g_2^* < g^*$ and $g_0 \leq g_2^*$, thus, $g_0 \leq g_2^* < g^*$, from which it can be judged that when $\alpha \neq \beta$, green supply chain coordination cannot be achieved.

(2) Assuming $\alpha = \beta$, by proposition 2, $g_1^* = g^*$, and by proposition 4, $g_2^* = g^*$. Because $g_2^* \geq g_0 \geq g_1^*$, it can be concluded that $g_0 = g_1^* = g_2^* = g^*$. If and only if $\alpha = \beta$, reward and punishment contracts can coordinate the supply chain.

## 5. Model Expansion Analysis

The above study assumes that the manufacturer in the green supply chain has a fair preference, and the coordination of the green supply chain when the channel strength of the manufacturer and the supplier is equivalent. However, the reality is that the bargaining power and discourse power of manufacturers and suppliers are different, which will lead to the channel strength being not quite equal. This is due to the enterprise's own resources

and external market environment, which will lead to different statuses of participants in the supply chain. According to Kahneman's prospect theory, the fairness preference will be affected by different reference points [41]. In order to further analyze the impact of fairness preference on the coordination of green supply chain, different reference points of fairness preferences are selected to further study. When the channel power of suppliers and manufacturers in green supply chain is unequal, the manufacturer's profit function is:

$$\mu_m(\pi) = (\alpha - \beta)\pi_m + \varepsilon\frac{\beta\pi}{2} = (\alpha - \beta)[(p - C_m)(a - bp + \gamma g) - T] + \varepsilon\frac{\beta}{2}\left[(p - C_m - C_s)(a - \mathrm{bp} + \gamma g) - \frac{1}{2}\delta g^2\right] \quad (40)$$

The fair reference coefficient is $\varepsilon$, which reflects the difference in channel power between suppliers and manufacturers. This paper studies a two-echelon supply chain consisting of suppliers and manufacturers. If the manufacturer is in a weak position and its equilibrium income is less than that of the supplier, the value range of $\varepsilon$ is $0 < \varepsilon < 1$.

**Proposition 6.** *In the green supply chain, when the dominant positions of suppliers and manufacturers are different, when $\alpha/\beta > 1$, the larger $\varepsilon$ is, the smaller the corresponding product price p is and the smaller the green degree g is. When $\alpha/\beta < 1$, the greater $\varepsilon$ is, the greater the corresponding product price p is and the greater the green degree g is.*

**Proof of Proposition 6.** When considering the different dominant position of manufacturers and suppliers in the green supply chain, from proposition 1 and proposition 3, $p$ and $g$ have the optimal solution and are unique. The optimal solution is:

$$p_3^* = \frac{2\gamma^2(\alpha - \beta)(C_m + w - \mu) + \beta\gamma^2(C_m + C_s) - \varepsilon\beta a\delta - \frac{2b\delta\beta\varepsilon}{2\alpha - 2\beta + \varepsilon\beta}\left[(\alpha - \beta)(C_m + w - \mu) + \frac{\beta}{2}\varepsilon(C_m + C_s)\right]}{2(\alpha - \beta)\gamma^2 + \beta\varepsilon\gamma^2 - 2b\beta\delta\varepsilon} \quad (41)$$

$$g_3^* = \frac{2\gamma b(\alpha - \beta)(C_m + w - \mu) + b\gamma\beta\varepsilon(C_m + C_s) - 2\gamma a(\alpha - \beta) - \beta a\gamma\epsilon}{2(\alpha - \beta)\gamma^2 + \beta\varepsilon\gamma^2 - 2b\beta\delta\varepsilon} \quad (42)$$

$\beta > 0$, divide the upper molecular denominator with $\beta$ and obtain:

$$g_3^* = \frac{2\gamma b\left(\frac{\alpha}{\beta} - 1\right)(C_m + w - \mu) + b\gamma\varepsilon(C_m + C_s) - 2\gamma a\left(\frac{\alpha}{\beta} - 1\right) - a\gamma\epsilon}{2\left(\frac{\alpha}{\beta} - 1\right)\gamma^2 + \varepsilon\gamma^2 - 2b\delta\varepsilon} \quad (43)$$

Then, $g_3^*$ is derived from $\varepsilon$ to obtain:

$$\frac{\partial g_3^*}{\partial \varepsilon} = \left(1 - \frac{\alpha}{\beta}\right)\gamma b\frac{\alpha\gamma^2(w - \mu - C_m) + 4\delta[a - b(C_m + w - \mu)]}{\left[2\left(\frac{\alpha}{\beta} - 1\right)\gamma^2 + \varepsilon\gamma^2 - 2b\delta\varepsilon\right]^2} \quad (44)$$

Because $w - \mu - C_m > 0$, and $a - b(C_m + w - \mu) > 0$. So when $\alpha/\beta > 1$, $\frac{\partial g_3^*}{\partial \varepsilon} < 0$, that is, under the same fairness preference of manufacturers and suppliers, the greater $\varepsilon$, the smaller the corresponding green degree $g$.

Similarly, the relationship between $p_3^*$ and $\varepsilon$ can be proved. □

**Proposition 7.** *When suppliers and manufacturers have different dominant positions in the green supply chain, and the degree of fairness preference is equal, the overall income of the manufacturer in the lead is greater than the overall income of the supplier in the lead. When the dominant position is equal, the overall revenue of the supply chain is the largest.*

**Proof of Proposition 7.** It is proved that the overall profit function $\pi(p, g)$ is a concave function of product price $p$ and product green degree $g$. When $\alpha = \beta$, the overall profit of the green supply chain is the largest. When $\alpha > \beta$, $\frac{\partial g_3^*}{\partial \varepsilon} < 0$. When $\alpha < \beta$, $\frac{\partial g_3^*}{\partial \varepsilon} > 0$.

Based on the above conclusions, when $\alpha < \beta$, $g$ and $\varepsilon$ are positively correlated, and $\pi(p,g)$ increases with the increase of $g$. Then, when $\alpha > \beta$, $g$ and $\varepsilon$ are negatively correlated, and $\pi(p,g)$ decreases with the increase of $g$, so $\varepsilon$ and $\pi(p,g)$ are positively correlated. Under the same fairness preference of manufacturers and suppliers, the overall revenue of green supply chain increases with the increase of $\varepsilon$, and when $\varepsilon = 1$, the overall revenue is the largest. □

## 6. Numerical Examples

In order to further discuss and check the model, numerical calculations and simulations are used for numerical research. In this section, two cases of different green degrees of products, provided by suppliers, are analyzed.

(1) If $g > g_0$, suppose the production is market demand, that is, $q = (a - bp + \gamma g)$, Simulate and assign the relevant parameters, $q = 550$, $\delta = 450$, $b = 10$, $\gamma = 18$, $C_m = 5$, $C_s = 15$, $\theta = 1.5$, $w = 20$, $g^* = 0.73$, $p^* = 38.15$ from (15) and (16). At this time, the overall profit of the green supply chain is the largest, with the maximum value of 3176.87. Further analysis of the impact of $\alpha/\beta$ value on the green supply chain is shown in Table 3 and Figure 1. It can be found that when $\alpha/\beta < 1$ and $\alpha/\beta > 1$, the overall profit of the green supply chain is not optimal. When and only when $\alpha/\beta = 1$, the overall profit of the green supply chain is maximized. When $\alpha/\beta < 1$, with the decrease of the $\alpha/\beta$ value, the product price and green degree are also decreasing, the manufacturer's profit is decreasing, the supplier's profit is increasing, the overall profit of the supply chain is decreasing and the supply chain coordination is not achieved. When $\alpha/\beta > 1$, with the increase of the $\alpha/\beta$ value, the product price and green degree are also increasing, the manufacturer's profit is increasing, the supplier's profit is decreasing, the overall profit of the supply chain is decreasing and the supply chain coordination is not achieved yet. Therefore, only when $\alpha = \beta$, that is, the manufacturer's attention to their own interests and fairness are equal, the reward and punishment contract mechanism of the green supply chain can achieve coordination.

**Table 3.** Effect of manufacturer's fairness preference on green supply chain when $g > g_0$.

| $\alpha/\beta$ | $p$ | $g$ | $q$ | $T$ | $\pi_s$ | $\pi_m$ | $\pi(p,g)$ |
|---|---|---|---|---|---|---|---|
| 0.60 | 24.72 | 0.25 | 307.21 | 6605.06 | 1983.29 | -546.47 | 1436.82 |
| 0.70 | 32.95 | 0.36 | 227.02 | 4880.91 | 1445.94 | 1464.71 | 2910.64 |
| 0.80 | 35.77 | 0.48 | 201.01 | 4321.68 | 1254.19 | 1862.85 | 3117.04 |
| 0.90 | 37.23 | 0.60 | 188.56 | 4053.94 | 1143.70 | 2023.30 | 3167.00 |
| 1.00 | 38.15 | 0.73 | 181.54 | 3903.01 | 1061.34 | 2115.53 | 3176.87 |
| 1.10 | 38.81 | 0.85 | 177.24 | 3810.66 | 989.21 | 2181.35 | 3170.57 |
| 1.20 | 39.31 | 0.98 | 174.51 | 3751.96 | 919.43 | 2235.14 | 3154.57 |
| 1.30 | 39.71 | 1.11 | 172.76 | 3714.42 | 847.89 | 2282.88 | 3130.77 |
| 1.40 | 40.06 | 1.24 | 171.68 | 3691.13 | 772.14 | 2327.46 | 3099.60 |
| 1.50 | 40.36 | 1.37 | 171.07 | 3677.94 | 690.53 | 2370.42 | 3060.95 |
| 1.60 | 40.63 | 1.50 | 170.80 | 3672.20 | 601.90 | 2412.63 | 3014.53 |
| 1.70 | 40.87 | 1.64 | 170.80 | 3672.17 | 505.28 | 2454.64 | 2959.92 |
| 1.80 | 41.10 | 1.78 | 171.01 | 3676.63 | 399.88 | 2496.80 | 2896.68 |
| 1.90 | 41.32 | 1.92 | 171.38 | 3684.72 | 284.99 | 2539.34 | 2824.34 |
| 2.00 | 41.52 | 2.06 | 171.90 | 3695.81 | 159.95 | 2582.46 | 2742.41 |
| 2.10 | 41.72 | 2.21 | 172.53 | 3709.44 | 24.12 | 2626.27 | 2650.39 |

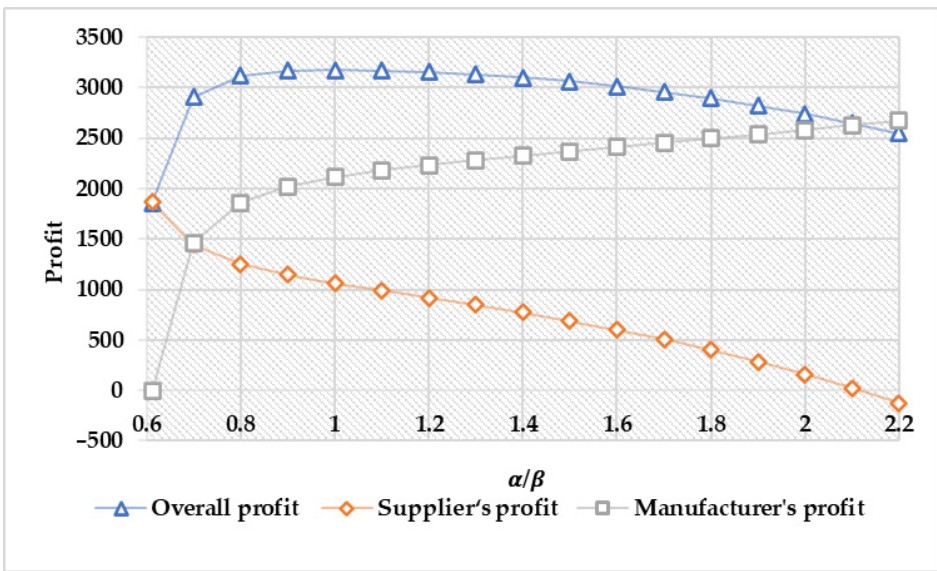

**Figure 1.** Relationship between overall profit and $\alpha/\beta$ when $g < g_0$.

(2) When $g < g_0$, because the green degree of the product is lower than the manufacturer's demand standard, it will have a certain impact on the market demand and reduce the market demand. Based on this, set $q = 500$, $\delta = 500$, $b = 10$, $\gamma = 20$, $C_m = 5$, $C_s = 15$, $\mu = 1$, $w = 20$, $g^* = 0.63$ and $p^* = 35.63$. At this time, the overall profit of the green supply chain is 2343.75. Compared with the equilibrium result when $g > g_0$, the green degree, prices and overall profits have all declined, which means that the green degree of the products supplied by suppliers will directly affect the green degree of the final product. Therefore, suppliers' strengthening of green innovation will increase product green degree and overall profit.

The influence of $\alpha/\beta$'s value on product price, the green degree, manufacturer profit, supplier profit and the overall profit of the green supply chain is further analyzed. See Table 4 and Figure 2 for details. The product price, green degree and manufacturer profit increase with the increase of $\alpha/\beta$, and the supplier profit decreases with the increase of $\alpha/\beta$. The overall profit of the green supply chain is optimal only when $\alpha/\beta = 1$, and the overall profit increases with the increase of $\alpha/\beta$ when $\alpha/\beta < 1$ and decreases with the increase of $\alpha/\beta$ when $\alpha/\beta > 1$. Only when $\alpha = \beta$, when manufacturers pay equal attention to their own interests and fairness, the reward and punishment contract can coordinate the green supply chain.

**Table 4.** Effect of manufacturer's fairness preference on green supply chain when $g < g_0$.

| $\alpha/\beta$ | $p$ | $g$ | $q$ | $T$ | $\pi_s$ | $\pi_m$ | $\pi(p,g)$ |
|---|---|---|---|---|---|---|---|
| 0.58 | 24.00 | 0.16 | 263.18 | 5000.40 | 1046.32 | 0.56 | 1046.87 |
| 0.60 | 27.19 | 0.19 | 231.85 | 4405.24 | 918.82 | 738.57 | 1657.39 |
| 0.70 | 32.29 | 0.29 | 182.93 | 3475.61 | 710.29 | 1516.95 | 2227.25 |
| 0.80 | 34.07 | 0.40 | 167.35 | 3179.64 | 629.07 | 1684.93 | 2314.00 |
| 0.90 | 35.01 | 0.51 | 160.12 | 3042.36 | 574.86 | 1763.35 | 2338.21 |
| 1.00 | 35.63 | 0.63 | 156.25 | 2968.75 | 527.34 | 1816.41 | 2343.75 |
| 1.10 | 36.07 | 0.74 | 154.06 | 2927.17 | 479.53 | 1859.96 | 2339.49 |
| 1.20 | 36.43 | 0.86 | 152.85 | 2904.06 | 428.23 | 1899.46 | 2327.69 |
| 1.30 | 36.72 | 0.97 | 152.24 | 2892.63 | 371.63 | 1937.20 | 2308.83 |
| 1.40 | 36.98 | 1.09 | 152.06 | 2889.13 | 308.58 | 1974.30 | 2282.87 |
| 1.50 | 37.22 | 1.22 | 152.17 | 2891.30 | 238.19 | 2011.34 | 2249.53 |
| 1.60 | 37.43 | 1.34 | 152.51 | 2897.73 | 159.74 | 2048.70 | 2208.43 |
| 1.70 | 37.64 | 1.47 | 153.02 | 2907.45 | 72.58 | 2086.58 | 2159.17 |
| 1.80 | 37.83 | 1.60 | 153.67 | 2919.81 | −23.87 | 2125.16 | 2101.29 |

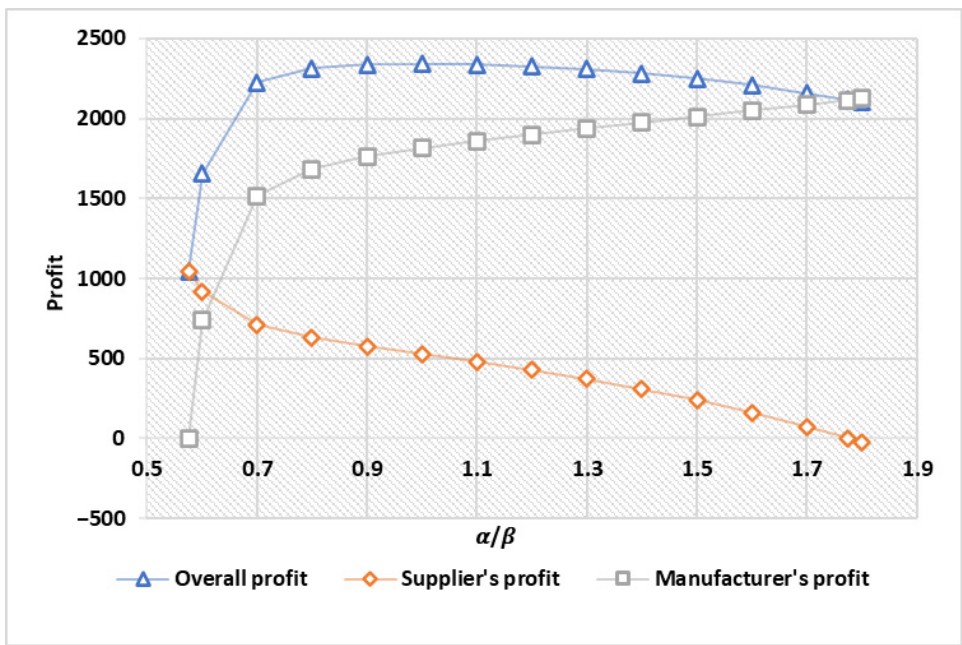

**Figure 2.** Relationship between overall profit and $\alpha/\beta$ when $g > g_0$.

Through the above analysis, it can be seen that in the reward–punishment contract with the green degree of the product provided by the supplier, the green degree of the product will increase with the decrease of the manufacturer's fairness preference. That is to say, the more the manufacturer attaches importance to its own interests, the more that the green degree of the product can be improved, so that the sales price of the product will also be improved. In the case of stable production, the profit of the manufacturer will also be improved. The improvement of the product green degree requires the green innovation input of the supplier, so that the cost of the supplier increases and the profit of the supplier decreases. The increase in the cost of green innovation leads to the decrease of the overall profit of the green supply chain.

(3) The numerical analysis above demonstrates that the channel power of suppliers and manufacturers in the green supply chain is equivalent. When the channel power is not equal, that is, when the supplier is stronger, perform a numerical simulation when $\varepsilon < 1$ and $\varepsilon = 0.3$, $\varepsilon = 0.5$, $\varepsilon = 0.7$, $\varepsilon = 0.9$, $\varepsilon = 1$. The results are shown in Figures 3 and 4.

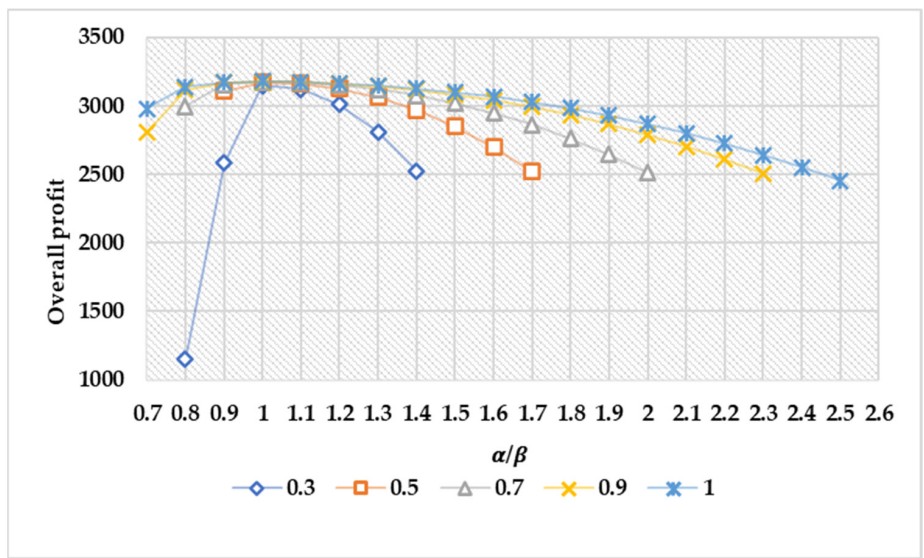

**Figure 3.** The relationship between the overall profit and $\alpha/\beta$ under different $\varepsilon$ values when $g > g_0$.

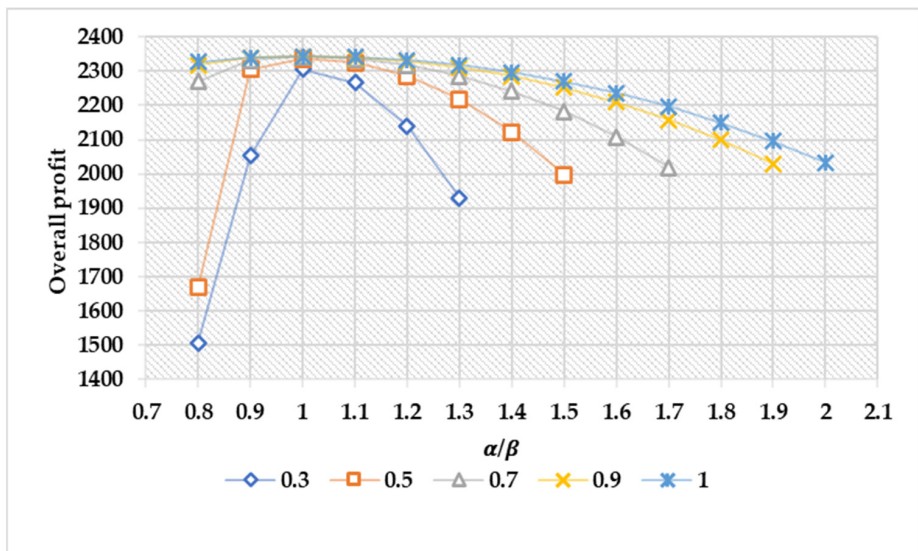

**Figure 4.** The relationship between the overall profit and $\alpha/\beta$ under different $\varepsilon$ values when $g < g_0$.

It can be seen from the figure that when the channel power of suppliers and manufacturers is not equal, the stronger the supplier is, the smaller the $\varepsilon$ is, the greater the sensitivity of the overall profit of the green supply chain to the change of the fairness preference $\alpha/\beta$ value is, and it is less than the overall profit of the green supply chain when $\varepsilon = 1$. This also just shows that the unequal channel power of the member companies of the supply chain damages the overall interests of the supply chain. This inequality will affect the win-win cooperation between the member companies of the supply chain. When the manufacturer has a fair preference, the manufacturer gives up part of the income, which is conducive to improving the overall income of the supply chain. Manufacturers should pay attention to fairness while paying attention to their own income. When the two are equally concerned, the overall profit can be maximized, thereby realizing the coordination of the green supply chain.

## 7. Managerial Insights

In the actual economy and society, the model provides decision-making references for supply chain members to cooperate in the process of green innovation investment. The purpose is to realize the profit of all parties in the green supply chain and promote the benign development of the green supply chain. The following are the recommendations for improving the green supply chain:

(1) This paper studies a green supply chain model of suppliers and manufacturers. Manufacturers should pay attention to fairness as well as their own benefits as it is crucial for a reliable green supply chain management system, which helps to improve product green degree and supply chain revenue.

(2) Managers should pay attention to fairness while paying attention to their own benefits. When the level of attention to the fairness and the benefits is equal, the overall profit maximization can be achieved.

(3) If the manufacturer's expected revenue of green innovation investment increases more than the expected revenue of the green suppliers, the manufacturer needs to increase incentives to offset the reduced revenue of suppliers. If the expected revenue increase of green manufacturers is less than the expected revenue increase of green suppliers, green manufacturers could use punishment mechanisms to increase their revenue.

(4) This model considers green innovation to increase profit by improving the green degree of the products. Managers improve product green degree through collaborative innovation, this research clearly shows variations in the profit depending on the green degree of the product.

(5) Managers of upstream and downstream enterprises in the supply chain should strengthen exchanges and cooperation. One channel member has considerable strength and dominant position, and the other members are willing to cooperate and require the establishment of interdependence.

## 8. Conclusions

The object of this paper is a two-stage, green supply chain system composed of a supplier and a manufacturer. When the manufacturer has a fair preference, the 'ERC' fair preference theory is used to construct a reward and punishment contract model with green degree as the standard. By analyzing the influence of different reward and punishment conditions and the difference of channel strength of each main body of the green supply chain on the green degree, price, supplier profit, manufacturer profit and overall profit of the whole supply chain, the conclusions are as follows:

(1) For manufacturers with fairness preference in the green supply chain, when the degree of attention to fairness is equal to the degree of attention to their own interests, the overall profit reaches the maximum. The equilibrium result at this time can realize the coordination of the reward and punishment contract on the green supply chain. When the degree of attention to fairness is greater than that of their own interests, the overall profit does not reach the maximum, but it will increase with the increase of the degree of fairness preference of manufacturers. When the degree of attention to fairness is less than that to their own interests, the overall profit deviates from the optimal value with the decrease of the degree of fairness preference of manufacturers.

(2) The coordination condition of reward and punishment contract in green supply chain based on manufacturer's fairness preference is related to the degree of fairness preference and has nothing to do with the channel strength of member enterprises in the supply chain.

(3) In the green supply chain with manufacturers' fair preference, the higher the green degree of the product, and the higher the price of the product, the manufacturer's income will increase with the increase of the green degree of the product and the supplier's income will decrease with the increase of the green degree of the product. Only when the manufacturer's attention to fairness and their own interests are equivalent is the overall profit is optimal. However, as manufacturers pay more attention to their own interests, although their profits are increasing, the overall profits are decreasing. In order to ensure the fairness of profit distribution, manufacturers prefer to sacrifice part of their own interests for a more equitable result.

(4) The difference in the channel power of member enterprises in the green supply chain will affect the overall profit of the supply chain. If the supplier dominates, and the stronger the dominant position is, the greater the impact on the overall profit of the green supply chain is. Only when the channel power of the supplier and the manufacturer is equal, the overall profit of the green supply chain is the largest.

This paper innovatively puts forward the reward and punishment contract based on the green degree reference standard under the condition of fairness preference and expands the research on the coordination of supply chain contracts with different fairness reference points when the statuses of the participants in the supply chain are different. The research in this paper supplements the existing research on green innovation decision-making of supply chains. Although this paper puts forward some opinions in green supply chain decision making, there are also some limitations. Firstly, this paper only focuses on the supply chain model composed of one supplier and one manufacturer. In the real competitive environment, the supply chain structure is complex, and this paper only considers the situation of information symmetry. In the future, we can continue to study the contract coordination between multiple supply chain members in the case of fairness preference and study the supply chain contract coordination problem considering fairness preference in the case of information asymmetry so that the model can be more in line with

the real decision-making environment and better reveal the phenomena and rules in the economic society.

**Author Contributions:** Data curation, M.J., D.C. and H.Y.; formal analysis, M.J., D.C. and H.Y.; methodology, M.J. and D.C.; project administration, M.J.; supervision, D.C.; writing—original draft, M.J.; writing—review and editing, D.C. All authors have read and agreed to the published version of the manuscript.

**Funding:** This research was funded by the National Natural Science Foundation of China: "Supply chain advertising decision optimization considering delay effect under different competition structures (72001059)".

**Institutional Review Board Statement:** Not applicable.

**Informed Consent Statement:** Not applicable.

**Data Availability Statement:** We have not reported data yet.

**Conflicts of Interest:** The authors declare no conflict of interest.

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
