# Peer review of "Research on Reward and Punishment Contract Model and Coordination of Green Supply Chain Based on Fairness Preference"

_sustainability, doi:10.3390/su13168749_

Round 1

Reviewer 1 Report

The purpose of the article is interesting because of the importance of the question raised in order to maximize the equity of the green supply chain.

The study has been raised from a theoretical basis for the proposal of a model based on the contributions of Chiu et al. (2010). The exposition of both the model and the basic assumptions on which it is based is quite clear, however, it is asserted that this paper improved the original model using the green grade standard, built a two-level green supply chain model made up of manufacturers and suppliers. Perhaps it would be convenient to simply present the variations made in the original cited model, in order to describe the methods used in the study, waiting for the validation by means of experimental study cases of the new proposed model before affirming that it constitutes a verifiable improvement.

The Bolton (1997) reference (page 4 line 176) appears mis-referenced in the text and has typographical errors in the references section. Something similar occurs in section 5 (page 8 line 330) when quoting the Kahneman prospect theory. Similarly, a typographical error has been detected in the citation [16]. Where it says Chiuc should put Chiu. A thorough review of all references is required to ensure that they are correct and comply with the publication format.

Sections 3 and 4 generate some confusion since they both have the same title “Improved model 'ERC'”. This makes it difficult to understand both sections and their relationship with the previous ones. It is necessary to review the organization of the sections so that the structure of the article is clear and does not confuse the reader.

In section 6 different numerical examples are presented to further discuss and check the model. Although the studied values ​​are listed in the corresponding tables, it is not clear whether the values ​​used correspond to a simulation to test the model or actual values. In case of being simulated values, perhaps it would be convenient to better justify their selection. In case of being real values, it would be convenient to better present which case studies it corresponds to.

In all Figures it is necessary to include the titles and units of both axes so that the content presented can be adequately evaluated.

Author Response

Thank you for your letter and for the reviewers’ comments concerning our manuscript entitled. Those comments are all valuable and very helpful for revising and improving our paper, as well as the important guiding significance to our researches. We have studied comments carefully and have made correction which we hope meet with approval. Revised portion are marked in the paper. The main corrections in the paper and the responds to the reviewer’s comments are as flowing:

  1. References:We have added some articles, proofread and revise carefully.
  2. The title of Sections 4 has been modified.
  3. The values of section 6 ​​is simulated assignment.
  4. We have added titles for both axes of all figures.

Reviewer 2 Report

attachment. 

Author Response

  1. We have carefully proofread and revised about manuscript.
  2. “ERC” is a theory of Equity, Reciprocity and Competition(In 1990s, Proposed by Gary E Bolton). In view of your opinions, we have deleted “ERC” in keywords.
  3. Sections 3 is “Improved model “ERC” ,and there is an elaboration on “ERC”, see page 6 line 282- 293.
  4. We have added research gaps, see page 4 line 194 to page 5 line 213.
  5. We have added few articles on Nash bargaining and coordination supply chain.
  6. The authors' contributions have been added.
  7. We have added practical implications of the research, see page 15 line 604- 609.
  8. 8.The limitations of study have been added, see page 15 line 610- 618.

Reviewer 3 Report

  1. The abstract and conclusions are very short.  The way of abstract writing is not perfect. The abstract should contain the details of the study and the findings in a very constructive way. The abbreviation should not be in the abstract. If needed, it can be started from the introduction onwards. The conclusions should be extended with significant findings and limitations. The applicability of the model should be explained.
  2. The research gap should be adequately explained.
  3. In the introduction, please rearrange/rewrite so that each authors’/most of the authors' contributions should be linked. Please try to maintain the literature sequentially.
  4. The introduction should be based on the exact research gap, and the literature review should be based on the specific keywords-based review, and finally, make an author's contribution table to show the novelty and effectiveness of the study.
  5. Please write proper managerial insights to show the industry managers' benefit from this research and compare this study with these studies "A supply chain model with service level constraints and strategies under uncertainty; Involvement of controllable lead time and variable demand for a smart manufacturing system under a supply chain management" theoretically and methodologically the applicability of the proposed research.
  6. Please write the significant findings in conclusions. Do not mention all assumptions which have been indicated within the model.
  7. What is the data source of the numerical experiment? Please mention that the data is from industry or literature, i.e., accurate data or artificial data.
  8. Make the graphical representation properly for the research from the sensitivity analysis table.

Author Response

  1. We have rewritten the abstract and the conclusion.
  2. We have added research gaps, see page 4 line 194 to page 5 line 213.
  3. We have rewritten the introduction.
  4. We have rewritten the literature review, but we apologize that due to time constraints, we did not add the author contribution table.
  5. We have added managerial insights of the research, see page 15 line 604- 609.
  6. We have added the significant findings in the conclusion.
  7. The data source is simulated assignment.
  8. Thank you for your suggestion. We will follow up on it.

Round 2

Reviewer 2 Report

  1.  Still, several typo mistakes and spacing problems can be seen in the revised version. 
  2. The authors' contribution table is missing. 
  3. The notation table is missing. Without this, it is difficult to follow the model.
  4. The authors haven't cited the references properly. Check References 1 ~6 and update the names. 
  5. Practical implications are missing. Add a new subsection before conclusions and highlight the insights. 

Author Response

Thank you for your letter and for the reviewers’ comments concerning our manuscript entitled. Those comments are all valuable and very helpful for revising and improving our paper, as well as the important guiding significance to our researches. We have studied comments carefully and have made correction which we hope meet with approval. Revised portion are marked in the paper. The main corrections in the paper and the responds to the reviewer’s comments are as flowing:

1.We have carefully proofread and revised about manuscript.
2.We have been added the table of authors' contribution 
3.We have been added the notation table.
4.We have been Checked the Reference.

5.We have been added the practical implications of study.

Reviewer 3 Report

The paper can be accepted for publication.

Author Response

Thank you for your letter and for the reviewers’ comments concerning our manuscript entitled. Those comments are all valuable and very helpful for revising and improving our paper, as well as the important guiding significance to our researches. 

Round 3

Reviewer 2 Report

No comments.